# An Improvement of the Cherenkov THz Generation Scheme Using Convex Silicon Prism-Lens Adapters

Galiya Kh. Kitaeva *[ID], Daniil A. Safronenkov [ID] and Natalia V. Starkova

Faculty of Physics, Lomonosov Moscow State University, 119991 Moscow, Russia;
safronenkov.da14@physics.msu.ru (D.A.S.); starkova.nv19@physics.msu.ru (N.V.S.)
* Correspondence: gkitaeva@physics.msu.ru

**Abstract:** The terahertz (THz) generation efficiency in the Cherenkov optical rectification scheme can be improved significantly if the silicon adaptor, mounted at the lateral surface of a nonlinear crystal, has a convex output surface with proper geometry. We demonstrate and compare with the results of direct experiments a method for theoretically modeling the angular distributions of the spectral power of THz radiation in the case of different Si adaptors, constructed by mounting plano-spherical lenses on a conventional flat Si prism. The applied method of theoretical modeling shows its usefulness in choosing the best Si adapter geometry.

**Keywords:** terahertz; optical rectification; Cherenkov THz generation; Si prism adaptor

## 1. Introduction

Electromagnetic radiation in the frequency range from 0.3 THz (or 0.1 THz) to 10 THz (or even up to 30 THz) is usually considered as a separate type of radiation ("THz radiation"), mostly due to the specific features of its interaction with matter and to its fairly weak degree of development compared to neighboring spectral ranges [1,2]. Since THz radiation is non-ionizing, it has a great potential for use in biophotonics and medicine [3], as well as in security systems [4]. THz radiation propagates without absorption through many dielectric materials [5], but has characteristic absorption lines in other substances, which is important for non-destructive testing and non-invasive diagnostics [6]. The practical application of THz radiation in various scientific and technological fields is of particular interest, but still requires more progress in creating suitable sources and sensitive receivers of THz-range electromagnetic radiation.

Many methods currently used to generate high-power pulses of THz radiation are based on the optical rectification (OR) of femtosecond laser pulses in an electro-optical crystal. The advantages of such schemes are explained not only by the possibility of generating high-amplitude THz fields, but also by the possibility of detecting the subsequent response of the irradiated objects by the means of terahertz time-domain spectroscopy set-ups [7]. Various materials, such as GaAs, GaSe, GaP, ZnTe, CdTe, organic crystals DAST, OH1, and others [8,9], are used to implement the nonlinear optical OR process. Due to their high non-linearity, Mg-doped congruent and stoichiometric LiNbO$_3$ (LN) crystals are among the best materials for these schemes [10]. Nevertheless, exactly phase-matched collinear interaction between optical and THz waves is impossible in bulk LN samples. The terahertz phase velocity ($v_{ph}^{THz}$)–optical group velocity ($v_{gr}^{pump}$) mismatch imposes a strong limitation on the coherence length of the crystal, which is about 10 µm [11]. Various approaches have been developed to circumvent this problem. In periodically poled LN crystals, the collinear THz generation is arranged with a high spectral power density within a narrow spectral band [12,13]. Scanning of the THz generation frequency is carried out by changing the domain poling period [14]. The most popular scheme for highly effective broadband THz generation uses femtosecond-duration pump pulses with a tilted pulse

front [15]. Record values of optical-terahertz conversion efficiencies, up to 3.7% in cryogenically cooled congruent lithium niobate and 1.7% at room temperature, were obtained using this method [10,16,17]. In this scheme, THz waves are generated within the whole range of the crystal transparency at the lower polariton branch below 1.5–2 THz [18].

No less broadband THz waves can be generated with a high efficiency without tilting the pump pulse fronts, but by using the pump beams tightly focused in a line [19]. In this case, OR takes place in a narrow layer just along the crystal lateral surface. The maximum THz energy at each frequency is generated at the Cherenkov angle $\theta_{cher}$, determined as

$$\theta_{cher} = arccos\left( v_{ph}^{THz} / v_{gr}^{pump} \right). \tag{1}$$

The most convenient way of extracting THz radiation from a crystal turned out to be through a silicon (Si) prism coupler mounted at the lateral crystal surface. First proposed for sideways THz generation in optical parametric oscillators [20], Si prism couplers are used now in various schemes for difference-frequency THz generation [21,22], THz generation by the means of OR, and even under THz wave detection in LN crystals [23,24]. Transverse spatial compression of the pump field over large distances inside the long nonlinear medium is carried out by focusing the pump beam into the crystal samples made in the form of thin-plate waveguides [23,25]. Special sandwich structures based on such waveguides are used to join the THz waves generated at Cherenkov angles in the opposite transverse directions [26]. The undoubted advantage of using a Si prism adapter is that all the waves of different THz frequencies, generated directly at Cherenkov angles in the main crystal [0, 0, 1] plane, escape out from the crystal in the same direction. However, the THz radiation emerges inside the crystal not only at a Cherenkov angle in the main plane, and the total THz beam has a large angular divergence. It occurs due to the permissibility of some small phase-group mismatch in the polar direction in the main plane, and due to a significant contribution from OR processes that take place in other planes oriented in the crystal at various azimuthal angles. A huge fraction of the THz radiation energy cannot be completely transferred from the crystal to the external space due to the effects of total internal reflection at the crystal–prism and prism–air boundaries. Moreover, after passing through the flat output surface of the Si prism, the transmitted THz waves propagate in a wide cone and cannot be easily focused on a detector or other spatially localized object. In recent years, the Cherenkov scheme has been substantially improved. In particular, different types of spatially shaped output Si coupler surfaces have been used [27–29], such as spherical and cone-shaped ones. These examples show how important choosing the optimal Si adapter geometry can be. However, no reasons have been given as to why each form of the adaptor was chosen, and how to find the best shape of the output adaptor surface. Certainly, any directional search for the optimal form of the Si coupler needs a general theoretical model that can predict the spatial characteristics of the radiated THz power based on the geometry of the adapter.

Modeling of the spatial field distribution outside a Si prism with a flat output surface was performed in [25], based on the direct solution of Maxwell's equations in spatial variables [25,26,30,31]. However, this method has not been applied to obtain the detailed angular distribution of THz radiation even from a flat prism, not to mention that it is too difficult to use it in the case of a Si prism with a curved output surface. Recently, we proposed another approach for modeling angular distributions based on integrating solutions of the nonlinear wave equations for individual spatial modes of the pump and THz radiation [32].

The main goal of this paper was to experimentally check and analyze the predictive power of our modeling method [32] for the adaptors fabricated by mounting silicon planospherical lenses on a flat Si prism. In its presented form, the developed approach enables calculating the power density of the THz waves initially generated in the crystal at the intersection of the pump beam and the lens axis. We analyze and compare the results of the

direct experimental measuring of the THz signals, obtained with a number of prism-lens adaptors, with calculations made specifically for these experimental schemes.

## 2. Experimental

Experiments were carried out on the THz time domain set-up shown schematically in Figure 1a. An Er$^{3+}$ laser with a 1560 nm wavelength, 70 MHz pulse repetition rate, 100 fs pulse duration, and an average power of 120 mW was used as a source for the pulsed optical pumping. The radiation from the laser passed through a Faraday isolator and a $\lambda/2$ plate to protect the laser from reflections and obtain the desired polarization. The laser beam was divided by a beam splitter into the probe beam (20% of radiation power) to pump a THz detector (PCA, the Menlo-System semiconductor photoconductive antenna) and the pump beam (~100 mW of the mean power) for THz generation in a Mg:LiNbO$_3$ crystal (with 7.1 mol.% of Mg) with a 1.23 cm length. The pump beam was focused to an input Z-surface of the crystal using a spherical lens with a 5 cm focal length into a circular spot of $w_p \sim 100$ μm waist width. The pump and THz waves were polarized in the crystal along its optical Z-axis. A flat prism from high-resistive Si with an acute angle $\delta = 40°$ was mounted at the lateral surface of the crystal (Figure 1b). Here, R is the radius of curvature of the lens, $D$ is its diameter, $l$ is the thickness, $x_0$ is the distance at which the optical axis of the lens intersects the pump beam on the side surface of the crystal, and $a$ is the distance from the center of curvature of the lens to the point with coordinates {$x_0$, 0, 0}. The OX axis is directed along the propagation of the pump beam, the OY axis is perpendicular to the crystal–prism interface, and the OZ axis is co-directional with the optical axis of the crystal. Three silicon plano-spherical lenses with different parameters (Table 1), each in turn, were attached to the flat exit surface of the prism to enhance the total power of the extracted THz radiation. Terahertz radiation was collected at a THz-detecting photoconductive antenna (PCA) by two parabolic mirrors with a focal length of 10 cm.

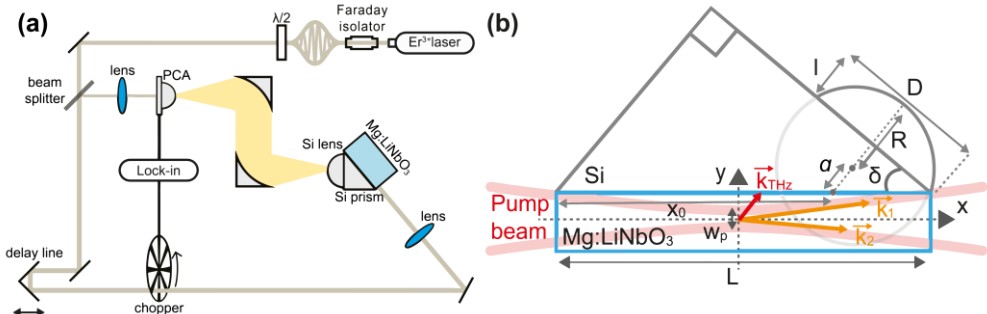

**Figure 1.** Sketches of the experimental set-up (**a**) and LN crystal with the prism and the lens overlay (**b**).

**Table 1.** Parameters of lenses used as overlays on the output surface of Si prism.

| Lens # | Curvature Radius R, mm | Diameter D, mm | Thickness l, mm |
|---|---|---|---|
| 1 | 6.0 | 10.4 | 3.0 |
| 2 | 6.0 | 11.5 | 4.3 |
| 3 | 5.0 | 10.0 | 6.3 |

The temporal profiles of the THz field, measured with each lens overlay and without any overlay on the prism, are shown in Figure 2a. Figure 2b demonstrates the corresponding THz power spectral distributions, obtained by the fast Fourier transform of the profiles. A spectral range of the detected THz radiation is limited by the area of PCA sensitivity. Both the amplitude and the spectral power density grow considerably when the lenses are applied. Comparing different temporal profiles, one can see that the best enhancement of

the THz field amplitude is observed when lens #2 is used as an overlay on the flat prism surface. This lens increases the peak to peak value in the output THz field by ~4.5 times.

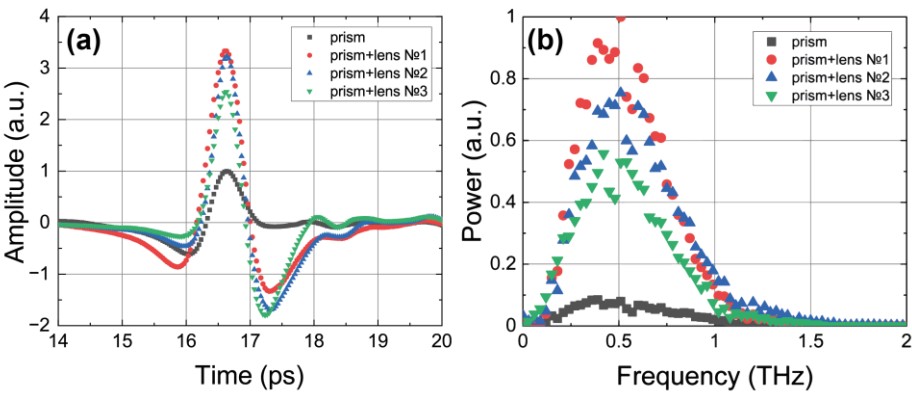

**Figure 2.** Temporal profiles of THz fields (**a**) and spectral power distributions (**b**), detected with a single flat prism and with the lens overlays mounted on the prism.

A spectral power density enhancement of ~12 times with lens #1 is achieved at a maximum of the spectral distribution, near 0.5 THz. Although the maximal spectral power densities differ noticeably in all the considered cases, the spectral widths remain approximately the same. However, a detailed study of the effect of adapter curvature on the THz generation bandwidth has to be carried out using a more broadband THz detector.

## 3. Theoretical Modeling

Following our previous theoretical approach [32], we analyze the THz generation process under the OR of the focused short-pulsed laser radiation as a continuum of the simultaneous difference frequency generation processes (DFG) between different plane-wave pump components. For example, the Gaussian spatial distribution of a pump component at some frequency $\omega_1$, with an amplitude $A_0$ and a beam waist $w_p$, is decomposed by plain waves with amplitudes

$$E_1(\mathbf{k}_{1\perp}) \sim A_0 w_p^2 e^{-\frac{w_p^2 \mathbf{k}_{1\perp}^2}{4}}. \tag{2}$$

Two different spectral components with frequencies $\omega_1$ and $\omega_2$ generate a THz wave at frequency $\omega_{THz} = \omega_1 - \omega_2$. For the corresponding wave vectors $\mathbf{k}_1, \mathbf{k}_2$, and $\mathbf{k}_{THz}$, the transverse phase-matching condition $\Delta\mathbf{k}_\perp \equiv \mathbf{k}_{1\perp} - \mathbf{k}_{2\perp} - \mathbf{k}_{THz\perp} = 0$ should be satisfied. The efficiency of the THz wave generation depends on the longitudinal phase mismatch $\Delta\mathbf{k}_{||}$, which is nonzero in bulk LN for these plane components. In a rather realistic case of low-gain approximation, expressions for the amplitudes of the generated THz waves are obtained and then the amplitudes of all elementary three-wave processes are summed up. As a result, the frequency and angular dependence of the total THz field amplitude at some frequency $\omega_{THz}$ are derived in the form:

$$A(\vartheta_{THz}, \varphi_{THz}, \omega_{THz}) \propto \frac{\omega_{THz}}{\sqrt{n_{THz}}} w_p^4 \chi^{(2)} \int_0^\pi d\vartheta_2 \sin\vartheta_2 \int_0^{2\pi} d\varphi_2 e^{-\frac{w_p^2}{4}[\mathbf{k}_{2\perp}^2 + (\mathbf{k}_{2\perp} + \mathbf{k}_{THz\perp})^2]} e^{-(\mu_{THz} + i\Delta k_{||})L/2} \frac{\sinh\left((\mu_{THz} + i\Delta k_{||})L/2\right)}{(\mu_{THz} + i\Delta k_{||})L/2}. \tag{3}$$

Here, $\Delta k_{||} \equiv \left|\Delta\mathbf{k}_{||}\right|, n_{THz}$ is a refractive index for the THz waves in the crystal (its value was taken for our calculations from [33] and the crystal dispersion parameters in the optical range were taken from [34]), $\mu_{THz} \equiv \frac{\alpha_{THz}}{2\cos\vartheta_{THz}}$ depends on the crystal absorption coefficient for the THz waves $\alpha_{THz}$ (taken for calculations from [18]) and polar angle of the THz wave propagation $\vartheta_{THz}$ with respect to the pump beam direction OX (Figure 3a), and the nonlinear coefficient $\chi^{(2)} \approx \chi\sqrt{1 + 3\cos^2\varphi_{THz}}$ depends on the azimuthal angle $\varphi_{THz}$ (measured from the OY axis directed perpendicular to the crystal/prism interface, see Figure 3a) in our case of an *eee*-type process in LN [35]. The

angular distribution of the THz power density at $\omega_{THz}$ inside the crystal can be found as $S(\vartheta_{THz}, \varphi_{THz}, \omega_{THz}) \propto |A(\vartheta_{THz}, \varphi_{THz}, \omega_{THz})|^2$. An example of this initial angular distribution of the THz radiation at frequency $\omega_{THz}/2\pi = 1$ THz, calculated for the pump beam waist $w_p = 100$ μm, is presented in Figure 3b. One can see a rather high initial angular divergence within the azimuthal angles.

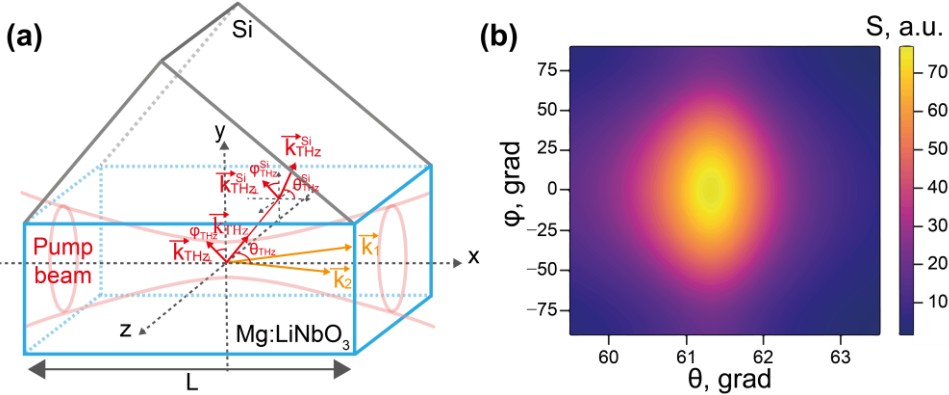

**Figure 3.** (**a**) The pump and THz waves inside LN crystal and Si adaptor. (**b**) Calculated angular distribution of the power density at 1 THz inside Mg:LiNbO$_3$ crystal.

To obtain the corresponding angular distribution inside the Si prism, for the THz radiation transmitted through the LN/Si surface, we had to decompose the as-generated THz waves into s- and p-polarized components and then use the Fresnel equations together with Snell's law. The same procedure was performed to obtain the final distribution in the air with flat or convex output surfaces of the Si lens-prism adaptor. As in [32], we took only one emitting point into consideration when calculating the adaptors with plano-spherical lens overlays. This point was located near the intersection of the crystal surface with the lens axis (it is marked by coordinates {$x_0$, 0, 0} in Figure 1b). The focusing or defocusing effect of the lens depends on the distance $a$ between this point and the center of the output lens sphere. This distance depends on the prism acute angle $\delta$, the lens diameter $D$, and its thickness $l$, $a = tg\delta D/2 - (R - l)$. Our calculations highlight the ratio between $a$ and the radius of the lens curvature $R$ as the main parameter that influences the spatial distribution of the THz radiation power at the output of the adaptor. The best spatial concentration of the output radiation can be obtained if the optimal ratio is achieved for all generating points in the crystal. In this regard, a conical shape of the output adaptor surface is preferable.

To compare the experimental and modeling results, the parameters of the lenses and prism for the numerical calculations were taken the same as described in Section 2. The obtained angular distributions of the output power density are presented in Figure 4a–d for equal angular intervals (5° for polar angles and 40° for azimuthal angles) around the directions of maximum intensity. Although the Si prism in the experiments had an acute angle slightly different from the optimal one, which should correspond to the Cherenkov angle for the LN crystal with 7.1 mol.% Mg doping, the THz waves emitted under $\theta_{cher}$ inside the crystal propagated in these central directions outside our adaptors. One can compare the color scales in Figure 4 and see how small the maximal intensity was in the case of the conventional prism (Figure 4a) in comparison to the adaptors provided by the lens overlays. The use of lens #2 led to an increase in the power density by almost two orders of magnitude (Figure 4c). At the same time, the angular divergence was minimal in this case. Obviously, the lens performance is determined by its ability to concentrate THz power in a small angular interval. This depends on the lens curvature and thickness. For example, lens #3 did not significantly improve the angular divergence after the flat prism, so the maximum intensity was not so high, as shown in Figure 4d.

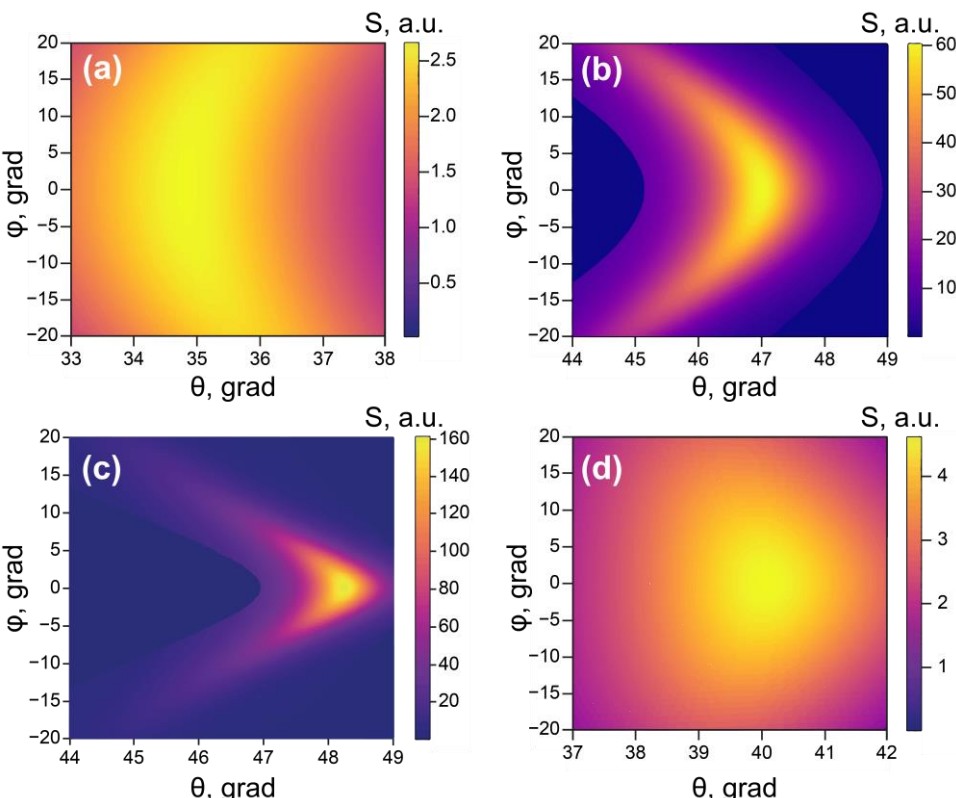

**Figure 4.** Angular distributions of the THz power density calculated for the schemes with (**a**) prism, (**b**) prism + lens #1, (**c**) prism + lens #2, and (**d**) prism + lens #3.

## 4. Discussion

First of all, let us compare the results of the numerical simulation with the results obtained in the previous work [31], where the same theoretical approach was described in detail and demonstrated using the example of several calculated distributions. The differences between these angular distributions and the ones presented here in Figures 3b and 4 are explained by the differences in the input parameters used under the numerical simulations. In [31], the acute angle of the prism was taken as optimal for the dispersion parameters of an LN crystal with 7.1 mol.% of Mg doping impurity when accounting for the transformation of the Cherenkov direction at the LN–Si boundary. In our experiments, a standard prism with δ = 40° was used, so we took into account this value in the simulations for this paper. We considered other ratios $a/R$ for the adaptors made with our lenses used in the experiments: 0.23 (lens #1), 0.52 (lens #2), and 1.10 (lens #3). In the case of a lensless prism, the polar angle of the maximal power density increased when the azimuthal angle shifted from its central value φ = 0 in both directions. However, at $a/R$ values typical for lenses #1 and #2, we observe the reverse behavior of the polar angle dependence. The direction of the polar angle variation change approximately near the $a/R$ value of lens #3. In the previous paper [31], the crystal absorption at the THz frequencies was taken into account only in several examples of calculations of the THz power density inside the crystal. Here, we account for the crystal absorption at the THz frequencies when calculating all the presented distributions and integral powers. That is why, in particular, the distribution of the power density is substantially wider in Figure 3b than in Figure 1b from [31]. Additionally, of course, taking into account the influence of absorption gives more realistic relationships between the THz radiation powers achievable using different lenses.

To make a comparison with the experimental data, we integrated the angular power densities over the angular range $-5^o \leq \varphi_{THz}^{air} \leq 5^o, -5^o \leq \vartheta_{THz}^{air} - \vartheta_{ph.m.}^{Si} \leq 5^o$, which is a realistic estimate for the detection aperture in our experimental set-up. Then, the numerical results obtained for different lens overlays were normalized to the power calculated for

the lens-free flat prism. The first columns in Table 2 show the calculation results obtained via this way for two frequencies, 0.5 THz and 1 THz. In the last two columns, there are experimental results taken from Figure 2b and normalized to the spectral power measured with the lens-free prism at corresponding frequencies.

**Table 2.** Comparison between the results of experiment and numerical calculations.

| Experimental Set-Up | P, a.u. Numerical Results | | P, a.u. Experimental Results | |
|---|---|---|---|---|
| | **0.5 THz** | **1.0 THz** | **0.5 THz** | **1.0 THz** |
| prism | 1 | 1 | 1 | 1 |
| prism + lens #1 | 25 | 22 | 12 | 6 |
| prism + lens #2 | 50 | 60 | 10 | 9 |
| prism + lens #3 | 2 | 2 | 7 | 3 |

As can be seen from Table 2, there was a fairly good quality-matching between the experiment and theory predictions. Lens #2 was the best in the most cases, lens #1 was slightly worse, and lens #3 lagged far behind. At the same time, the difference in the lens performance was not so much pronounced in the experiment as in the calculations. This was expected due to the fact that, in our calculation approach, we could consider only one THz-emitting point in the crystal. This point was located at the lens axis and the THz generation from it should be most sensitive to the lens characteristics. However, Table 2 shows that, despite this limitation of the theoretical calculation, the applied modeling approach was able to correctly indicate the best of the lenses in a real experiment.

As was mentioned above, for other generating points in the crystal, taken at distances from the crystal input surface longer or shorter than $x_0$, the optimal lens ratios $a/R$ could be not the same. At the same time, our calculations showed that the most pronounced effect of the optimal convex adaptor surface appeared due to the compensation of the widest azimuthal angular divergence of the THz waves that were initially emitted in the crystal. This indicates possible advantages of output adaptor surfaces with conical shapes. Although such a surface may not be as effective at polar angle collimation, it can significantly reduce the azimuthal divergence of the THz beams emitted along the entire length of the crystal. A bright line-shaped THz beam obtained this way can be further controlled using the necessary THz optical elements.

## 5. Conclusions

Summing up, we numerically and experimentally studied the effects of applying convex Si adaptors with spherical output surfaces on a nonlinear $Mg:LiNbO_3$ crystal under Cherenkov sideways THz generation. The numerical calculations of the angular power distributions at different THz frequencies were performed using a method based on integrating solutions of nonlinear wave equations for the individual spatial modes of the pump and THz radiation inside the crystal and accounting for further transformation of the THz wave components at the input and output adaptor boundaries. It was shown both theoretically and experimentally that the imposition of a plano-spherical lens on the output surface of a flat Si prism can lead to a significant increase in generation efficiency and helps to collimate the output THz radiation. The applied method of theoretical modeling showed its usefulness for choosing the best geometry of the output surface of a Si adapter.

**Author Contributions:** Conceptualization and writing, G.K.K.; experiment, visualization, writing—figures and editing D.A.S.; programming and calculations, N.V.S. All authors have read and agreed to the published version of the manuscript.

**Funding:** This research was funded by the Russian Science Foundation, grant number 22-12-00055.

**Institutional Review Board Statement:** Not applicable.

**Informed Consent Statement:** Not applicable.

**Data Availability Statement:** The main parts of the data underlying the results presented in this paper are available within the article; those not publicly available at this time and the more detailed parts may be obtained from the authors upon reasonable request.

**Acknowledgments:** D.A.S. expresses his gratitude to the Fund of Theoretical Physics and Mathematics Development "BASIS".

**Conflicts of Interest:** The authors declare no conflict of interest.

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
