# Peer review of "An Improvement of the Cherenkov THz Generation Scheme Using Convex Silicon Prism-Lens Adapters"

_photonics, doi:10.3390/photonics10101145_

Round 1

Reviewer 1 Report

The work is a sequel of the previous paper by the same group (Ref. 11), where combining of a plano-spherical Si lens with a Si-prism coupler was proposed to improve terahertz emission from a lithium niobate crystal excited by a femtosecond laser. The current paper is enriched by a comparison of theoretical and experimental results. The paper contains new (experimental) results and may be recommended for publication. However, the authors should clarify and improve the following issues.

There is a controversy between the theoretical results of the current paper and Ref. 11. In particular, in Fig. 3(b) the distribution of the power density over \theta is substantially wider than in the corresponding Fig. 1(b) of Ref. 11. Moreover, Figs. 4(b) and (c) look reversed with respect to the corresponding Figs. 4(b)-(d).

In Fig. 2(b), it would be more instructive to plot spectra normalized to unity. This would allow one to compare the spectrum bandwidths.

Reviewer 2 Report

See the attachments.

Pretty good.

Reviewer 3 Report

After careful reading the manuscript, authors are well described their results and discussion in crispy and clear way. I hope this short article is seems interesting to the readers. I hope this article is suitable for publication in Photonics.  

Author Response

We are much grateful to Reviewer for the kind kind attention paid to the manuscript and such a high assessment of our work.

Round 2

Reviewer 1 Report

The authors properly addressed all my concerns. I recommend the paper for publication.